# The Effect of Oxidant Hypochlorous Acid on Platelet Aggregation and Dityrosine Concentration in Chronic Heart Failure Patients and Healthy Controls

**DOI:** 10.3390/medicina55050198

**Published:** 2019-05-23

**Authors:** Aušra Mongirdienė, Jolanta Laukaitienė, Vilius Skipskis, Artūras Kašauskas

**Affiliations:** 1Department of Biochemistry, Medical Academy, Lithuanian University of Health Sciences, 44307 Kaunas, Lithuania; jolanta_laukaitiene@yahoo.co.uk (J.L.); Arturas.Kasauskas@lsmuni.lt (A.K.); 2Department of Cardiology, Hospital of Lithuanian University of Health Sciences, 44307 Kaunas, Lithuania; 3Laboratory of Molecular Cardiology, Institute of Cardiology, Lithuanian University of Health Sciences, 44307 Kaunas, Lithuania; skipskis@gmail.com

**Keywords:** platelet, heart failure, oxidant, dityrosine, hypochlorous acid

## Abstract

*Background and objective*: One of the reasons for thrombosis in chronic heart failure (CHF) might be reactive forms of oxygen activating platelets. The aim of this study was to evaluate the effect of oxidant hypochlorous acid (HOCl) on platelet aggregation and dityrosine concentration in CHF patients and healthy controls. *Materials and Methods:* CHF patients (*n* = 67) and healthy (*n* = 31) were investigated. Heart echoscopy, 6-min walking test, complete blood count, platelet aggregation, and dityrosine concentration were performed. Platelet aggregation and dityrosine concentration were measured in plasma samples after incubation with different HOCl concentrations (0.15, 0.0778, and 0.0389 mmol/L). *Results:* Platelet aggregation without oxidant was lower (*p* = 0.049) in CHF patients than in controls. The spontaneous platelet aggregation with oxidant added was higher in CHF patients (*p* = 0.004). Dityrosine concentration was also higher (*p* = 0.032) in CHF patients. Platelet aggregation was the highest in samples with the highest oxidant concentration in both healthy controls (*p* = 0.0006) and in CHF patients (*p* = 0.036). Platelet aggregation was higher in NYHA III group in comparison to NYHA II group (*p* = 0.0014). Concentration of dityrosine was significantly higher in CHF samples (*p* = 0.032). The highest concentration of dityrosine was obtained in NYHA IV group samples (*p* < 0.05). Intensity of platelet aggregation, analyzed with ADP, was correlated with LV EF (r = 0.42, *p* = 0.007). Dityrosine concentration was correlated with NYHA functional class (r = 0.27, *p* < 0.05). *Conclusions:* The increase in platelet aggregation in CHF and healthy controls shows the oxidant effect on platelets. The increase in dityrosine concentration in higher NYHA functional classes shows a higher oxidative stress in patients with worse condition.

## 1. Introduction

Thrombotic complications occur in 11–44% of chronic heart failure (CHF) patients [1]. One of the reasons for this might be reactive forms of O_2_, which have been proven to be related to the prognosis of cardiovascular diseases [2]. Activated neutrophils producing more myeloperoxidase have been identified in CHF patients. Myeloperoxidase catalyzes the synthesis of hypochlorous acid (HOCl) in the blood, which is subsequently transformed into superoxide anions and other reactive forms of O_2_. Superoxide anions have been proven to take a role in pathogenesis of cardiovascular diseases by oxidizing plasma proteins, including fibrinogen [3,4] and lipoproteins [5,6], and by activating platelets [3,7,8,9,10,11]. Oxidized fibrinogen has been found to interact with thrombin and to form thrombus more rapidly [12]. Chronic activation of platelets is related to an increased risk of capillary thrombosis [13]. Fibrinogen is one of the plasma proteins taking part in blood clotting process, which is affected mostly by oxidants [14]. Oxidized fibrinogen becomes nitrogenized fibrinogen, which has been shown to be present in higher concentrations in chronic heart diseases (CHD) [15]. Subsequently, the higher concentration of nitrogenized fibrinogen might serve as a reason for thrombotic complications [16]. Nitrogenized fibrinogen contains dityrosine groups. So dityrosine concentration in plasma represents oxidized protein amount [4].

Data on interaction of oxidized fibrinogen with platelets are controversial: some authors have found that it stimulates clot formation [12,15,16], while the others found it suppressing [14]. Data by various authors on oxidant impact on platelets also differ; some authors’ results show platelets to be activated by oxidants [8,13], while others have proved platelet suppression [7]. Thus, further studies are needed to explain those interactions.

Additionally, it is important to analyze the impact of oxidants on platelet aggregation from the perspective of therapeutic approach, i.e., platelet aggregation might be suppressed by antiaggregant and antioxidative agents thus avoiding bleeding complications.

The aim of this pilot study was to evaluate the effect of HOCl on platelet aggregation in CHF patients and healthy controls. To achieve this, we set five tasks: (1) to evaluate the differences of HOCl effect on the intensity of platelet aggregation between CHF patients and healthy controls; (2) to estimate the differences of the impact of various HOCl concentrations on platelet aggregation between CHF patients and healthy controls; (3) to evaluate the differences of dityrosine concentrations between samples from CHF patients and healthy controls with different oxidant concentrations; (4) to analyze the differences of platelet aggregation and dityrosine concentration between different CHF functional classes; and (5) to determine a correlation between the intensity of platelet aggregation and the other analyzed parameters.

## 2. Material and Methods

### 2.1. Patients and Healthy Controls

All the investigations were approved and conducted in accordance with the guidelines of the local Bioethics Committee and adhered to the principles of the Declaration of Helsinki and Title 45, U.S. Code of Federal Regulations, Part 46, Protection of Human Subjects (revised November 13, 2001, effective December 13, 2001). The study was approved by the Regional Bioethics Committee at the Lithuanian University of Health Sciences (No. BE-2-102). All the patients enrolled have signed the informed consent to participate in the study. CHF patients with reduced ejection fraction (systolic heart failure class II‒IV according to the NYHA (New York Heart Association)) from the Department of Cardiology, Hospital of Lithuanian University of Health Sciences and healthy controls, who had not been using any antiaggregants during the last 2 weeks and experienced no other factors affecting platelet aggregation, were included in the study. Healthy controls were enrolled in the study if they had no clinical history of coronary artery disease, no known peripheral artery disease, or history of symptoms suggestive of angina pectoris or heart failure. Also persons with autoimmune diseases, active and chronic infections, or neoplastic conditions were excluded. The diagnosis of CHF was made following the guidelines for the diagnostics and treatment of heart failure approved by the European Society of Cardiology [17]. The inclusion criteria were as follows: 47–78 years of age; agreement to participate in the study; class II‒IV heart failure according to the NYHA with ischemic, dilated, or hypertensive cardiomyopathy. The clinical state was considered stable if there were no changes in functional class according to the NYHA; the same medications used during the past 3–4 weeks; there were no new heart failure symptoms; and no antiaggregant was prescribed to avoid bleeding complications. Table 1 shows the characteristics of the patients and controls.

### 2.2. Tests and Blood Sampling

The heart echoscopy (TEE) and 6-min walking test were performed, and complete blood count, platelet aggregation, and dityrosine concentration were determined. Blood samples for the complete blood count testing were taken from the forearm vein with a 20 G needle into 4.5 mL vacuum tubes with EDTA (ethylendiamintetraacetic acid) and were put into hematological analyzer COULTER LH 780 (USA, Brea, CA, USA). In order to investigate platelet aggregation, blood samples were taken from the forearm vein into 5 mL vacuum tubes with 3.8% sodium citrate. In order to prepare platelet rich plasma, the blood was centrifuged at 100× *g* for 15 min at room temperature. Platelet poor plasma was obtained when the rest of blood was centrifuged at 1000× *g* for 30 min. Platelet aggregation was investigated in platelet rich plasma using the aggregometer (Chrono-Log, Havertown, PA, USA) by the standard Born method [18]. ADP (3.8 mmol/L (the final working concentration), Chrono-log P/N 384) was used for aggregation induction. Spontaneous aggregation (SP) was registered without using an inductor. Dityrosine concentration in all samples from each participant was measured using spectrofluorescent method (spectrofluorometer Perkin Elmer LS-55, ex 310, em 410, Perkin Elmer Life Sciences, Buckinghamshire, UK) in relative units of fluorescence.

### 2.3. HOCl Effect on Platelet Aggregation

In order to evaluate the differences of HOCl (Sigma, Munchen, Germany) effect at various concentrations on platelet aggregation, the platelet rich plasma of CHF patients and healthy controls was investigated. Different plasma samples were prepared from every person’s plasma, using 10 μL of saline solutions with different HOCl concentrations: (1) 0.15 mmol/L, (2) 0.0778 mmol/L, and (3) 0.0389 mmol/L (these were the final HOCl concentrations), and the same amount of saline in control tests. The samples were incubated for 10 min at temperature 37 °C, then platelet aggregation intensity (%) was evaluated by using ADP. The intensity of spontaneous aggregation was evaluated in two additional samples—one with 10 μL of 0.15 mmol/L HOCl and the other with 10 μL of saline as presented in Figure 1.

In our previously work, we have established that platelet aggregation significantly decreased big HOCl concentrations (from 2.11–43.4 mmol/L final concentrations). Therefore, very low and low concentrations of HOCl were chosen in this work (from 0.00389–0.15 mmol/L final concentrations).

### 2.4. Statistical Analysis

Statistical analysis was performed using SPSS 8.0, STATISTICA 7 and EXEL software. Normality was assessed with the Kolmogorov-Smirnov and Shapiro-Wilk tests. Data were found to be normally distributed, and differences between groups were evaluated by the ANOVA test. Comparisons between categorical variables were made using the chi-square test. Correlations of measured readings were evaluated using the Pearson correlation coefficient. Data are expressed as mean ± standard deviation (SD). A two-tailed *p* < 0.05 was considered significant.

## 3. Results

### 3.1. Clinical Characteristics of Patients and Controls

The analyzed left ventricular ejection fraction (LV EF) was found to differ significantly between patients of NYHA II and NYHA IV, NYHA III and NYHA IV, and between CHF patients and healthy controls (*p* < 0.001, *p* < 0.001, and *p* < 0.001, respectively). The 6-min walking test results were different between NYHA patients of different classes and healthy controls (*p* < 0.001). The patients of NYHA II and NYHA III were older than NYHA IV and healthy controls (*p* = 0.024). Platelet count was higher in healthy controls (*p* = 0.033). Mean platelet volume (MPV) was higher in CHF patients than in healthy controls as shown in Table 1.

The differences in medication used, heart rhythm disturbances present, renal failure, dyslipidemia, and CHD cases in CHF group were not statistically significant. It was observed that platelet count (PLT) did not differ between the CHF and healthy control groups.

### 3.2. The Differences of Platelet Aggregation and Dityrosine Concentration between CHF and Control

Platelet aggregation and dityrosine concentration values were found to be significantly different in the CHF and control groups. They are presented in Table 2 and Table 3 and Figure 2.

Values of platelet aggregation, investigated with ADP, in samples with oxidant added were significantly lower in the CHF group compared to the control group (Table 2). No reliable differences in platelet aggregation in samples with saline added were obtained between the CHF and control groups, although mean values in the control group were found to be higher. The results of spontaneous aggregation without the oxidant added were higher in the CHF group (3.45 ± 2.88 in control and 3.74 ± 2.57 in CHF) but the difference was not significant. Spontaneous aggregation in samples with the oxidant added was also more intense in the CHF group (6.49 ± 3.85 in CHF, 4.94 ± 3.17 in control, *p* = 0.004) (Figure 1).

Dityrosine concentration was significantly higher in the CHF group’s samples with just saline (*p* = 0.049) and in samples with the lowest concentration of the oxidant added in comparison with samples with the other oxidant concentrations (*p* = 0.032). Dityrosine concentration in samples with the highest concentration of oxidant added was significantly higher in the healthy controls group (*p* = 0.025). Dityrosine concentration in samples with the average oxidant concentration did not differ between CHF patients and healthy controls (Table 3). With the increase of oxidant concentration, dityrosine concentration increased in samples from CHF patients and healthy controls with the exception of the average HOCl concentration (0.0778 mmol/L) in CHF patients.

With the increase of oxidant concentration, platelet aggregation increased in both groups with the exception of the average HOCl concentration (Table 2). In the latter concentration of oxidant, an increase of platelet aggregation in the CHF and control groups was not significant. It is important to note that in samples with the highest and the lowest amounts of the oxidant, platelet aggregation increased both in the CHF and control groups in comparison with platelet aggregation without the oxidant (Table 2). Besides, in samples with the lowest concentration of the oxidant added, platelet aggregation was significantly higher compared to the samples with the average concentration of the oxidant (*p* < 0.00001 in the control group and *p* = 0.654 in the CHF group), as well as it was significantly higher in samples with the highest oxidant concentration compared to the average oxidant concentration samples (<0.001 in the control group and *p=* 0.036 in the CHF group). Platelet aggregation in samples without the oxidant added was lower in the CHF and control groups compared to samples with the oxidant (*p* < 0.001 in the control group and *p* = 0.02 in the CHF group).

We found that in samples with the highest and the lowest concentration of the oxidant added, the increase of platelet aggregation compared to the one in samples without the oxidant did not differ significantly in the CHF and control groups (Table 2).

### 3.3. The Differences of Platelet Aggregation and Dityrosine Concentration Between Chf Subgroups

Analysis of platelet aggregation and dityrosine concentration differences in CHF subgroups according to NYHA functional class was performed. We found that results of ADP-induced platelet aggregation were significantly higher in NYHA III subgroup compared to NYHA II subgroup, and significantly lower in NYHA IV subgroup compared to NYHA II subgroup. The results of ADP-induced platelet aggregation were higher in NYHA III subgroup than in NYHA IV subgroup, although the difference was not significant (Table 4). Spontaneous platelet aggregation with the oxidant added was significantly higher in NYHA II subgroup compared to NYHA IV subgroup (7.67 ± 4.09 versus 4.44 ± 3.13, *p* = 0.019), and significantly lower in NYHA IV subgroup compared to NYHA III subgroup (4.44 ± 3.13 versus 7.29 ± 3.71, *p* = 0.047). There was no significant difference in spontaneous aggregation with the oxidant added between NYHA II and III subgroups. It should be noted that in all CHF subgroups, spontaneous aggregation was higher in samples with the oxidant added, compared to the ones without the oxidant (NYHA II: 7.67 ± 4.09 versus 4.0 ± 3.38; NYHA III: 7.29 ± 3.71 versus 4.25 ± 2.27; NYHA IV: 4.44 ± 3.13 versus 2.83 ± 2.01, *p* = 0.04). Spontaneous aggregation without the oxidant in different CHF groups was not significantly variable (*p* = 0.39).

Summarizing the mentioned, ADP-induced platelet aggregation results were significantly higher in NYHA III subgroup compared to NYHA II subgroup and significantly lower in NYHA IV subgroup compared to NYHA II subgroup. With the advance of CHF severity, spontaneous platelet aggregation with the oxidant added decreased significantly (*p* = 0.047).

Dityrosine concentration was significantly higher in CHF group’s samples with the highest concentration of the oxidant in comparison with the lowest one (*p* = 0.032), as shown in Table 5. Dityrosine concentration was the highest in NYHA IV samples (*p* = 0.05). Significant difference in dityrosine concentration was discovered between NYHA II and NYHA IV samples, except the lowest oxidant concentration (*p* = 0.680). Dityrosine concentration was higher in NYHA IV samples with the oxidant, compared to NYHA III samples (*p* = 0.002), except the samples with saline and the lowest oxidant concentration.

### 3.4. Correlations among Platelet Aggregation, Dityrosine Concentration, LVEF, and NYHA Functional Class

A weak correlation was obtained among spontaneous aggregation with the oxidant added and person’s age (r = 0.25, *p* = 0.018) and CHF severity (NYHA functional classes (r = −0.329, *p* = 0.045)). Platelet aggregation with ADP and with or without oxidants data weakly and moderately correlated with LV EF (r = 0.31, *p* = 0.007; r = 0.4, *p* < 0.001; r = 0.34, *p* = 0.0101; and r = 0.48, *p* < 0.00001 in ADP–saline, ADP1, ADP2, and ADP3 samples, respectively). A weak negative correlation was estimated between a spontaneous aggregation variable and dityrosine concentration in various samples: TRADP 1 (r = −0.2, *p* = 0.041), TRADP3 (r = −0.21, *p* = 0.047), TRSP1 (r = −0.25, *p* = 0.019), and TRSP-saline (r = −0.22, *p* = 0.042) (TR, dityrosine concentration; SP, samples for spontaneous aggregation investigation; ADP, samples for aggregation investigation with ADP (adenosine diphosphate); 1,2,3, different HOCl concentrations).

There were also reliable weak and moderate correlations between renal failure and dityrosine concentration (r = 0.27, r = 0.45, r = 0.39, r = 0.36, and r = 0.46 in samples ADP–saline, ADP1, ADP2, SP1 and SP-saline, respectively, *p* = 0.045). NYHA functional class correlated with dityrosine concentration in samples without ADP and the oxidant (r = 0.272, *p* = 0.045), as well as with the intensity of spontaneous aggregation in samples with the oxidant added (r = −0.33, *p* = 0.045).

Summarizing correlations of the parameters measured, the correlation between LV EF and platelet aggregation data should be noted, as well as the positive correlation between spontaneous aggregation and age and the negative correlation between spontaneous aggregation and dityrosine concentration. NYHA functional class correlated with dityrosine concentration in samples without ADP and the oxidant.

## 4. Discussion

Our result revealed significantly lower platelet aggregation with ADP in CHF group compared to healthy controls. Higher spontaneous aggregation was observed in the CHF group (*p* = 0.004 and *p* = 0.39 with and without the oxidant, respectively). Higher platelet aggregation with ADP in the CHF group is not consistent with our and other researchers’ results. In a source summarizing research data [19], this fact is explained in two ways. First, ADP-induced platelet aggregation (5 μmol) was higher in CHF compared with healthy individuals suggesting the relationship among endothelium, leukocytes, and platelets via adhesion proteins due to inflammatory response. Therefore, levels of PECAM-1, ICAM, VCAM, P-selectin, osteonectin, and PF4 were significantly higher in the CHF group than in the control group. Second, it was found that platelets in healthy persons secrete sialic acid, which is one of the compounds maintaining platelets in an inactive state. Lower platelet sialic acid levels were reported in CHF patients and it was concluded that this could be a contributing factor for the higher aggregability of the platelets. It should be noted that in experiments described in this article, the authors used 5 μmol ADP to induce platelet aggregation. In our experiments, we used a lower ADP concentration, i.e., 3.8 μmol. Despite this, there are other studies with the findings of higher platelet aggregation with ADP in healthy controls compared to CHF patients and of similar spontaneous aggregation in both groups [20] that are consistent with ours. We could not find any other data in literature about the differences between spontaneous aggregation in CHF patients and healthy persons. It should be noted that literature sources stating higher platelet aggregation in CHF patients give no data about functional class of the patients and the medication used. The latest research data show that standard CHF treatment reduces platelet aggregation significantly [21,22]. Thus, we think we are contributing to the work of previous researchers. Higher platelet aggregation in CHF patients could be observed in NYHA I functional class undergoing a non-standard treatment.

We also found that platelet aggregation with ADP was significantly higher in NYHA III group compared to NYHA II group, and significantly lower in NYHA IV group compared to NYHA II group. Spontaneous aggregation with the oxidant decreased significantly with CHF (*p* = 0.004). Spontaneous aggregation without the oxidant was higher in NYHA III group compared to NYHA IV (*p* = 0.042). Thus, platelet aggregation with ADP and spontaneous aggregation in the CHF subgroups was as follows: NYHA III, NYHA II, and NYHA IV. It may be supposed that the differences in platelet aggregation intensity in the CHF subgroups might be influenced by the platelet count or mean platelet volume. The platelet count in healthy controls was higher compared to the CHF group. However, we did not observe significant differences in the platelet count and MPV in the CHF subgroups. Also, there was no correlation found among the platelet count, MPV, and intensity of platelet aggregation.

In order to explain platelet aggregation differences among the CHF subgroups, we also analyzed a possible effect of CHF medication. There is literature data [21,22] stating that β-adrenoblockers used separately decrease platelet aggregation, and that angiotensin converting enzyme inhibitors reduce P-selectin expression as well. Platelet activity marker (CD40L, P-selectin, CD63P, MPV, and MPC) concentration in blood was significantly reduced by standard CHF treatment but the medication used for treatment in different CHF subgroups did not differ in our study. Thus, platelet aggregation differences among the CHF subgroups cannot be explained by our results.

Based on significant differences in plasma vWf levels between NYHA II and NYHA III subgroups (plasma vWf levels were higher in the functional class III than in the class II), it was determined that endothelial function was poorer in NYHA III subgroup compared to NYHA II patients [18]. In authors’ opinion, these parameter changes in CHF patients were due to the response to cytokine hyperproduction. More intense platelet aggregation in NYHA III subgroup might be due to endothelial dysfunction. In NYHA IV subgroup, even more compromised endothelial function might be expected, but platelet aggregation in this subgroup was significantly lower, compared to NYHA II. These differences cannot be explained by literature data or by our results.

The results of researches on platelet aggregation differences in the CHF subgroups are scarce and controversial. Some authors [22,23] have not found any association between platelet activity and CHF severity. Other researchers [20] also have not found any significant differences among NYHA subgroups, though the values of platelet aggregation they presented were higher in the lower LV EF (≤45%) subgroup; however, no information on ADP concentration used in this study is presented. The authors have not found any spontaneous aggregation differences between CHF patients and healthy controls among the CHF subgroups.

We found a weak correlation between NYHA functional class and dityrosine concentration in samples without ADP and the oxidant. In order to explain varying platelet aggregation in CHF subgroups, further studies are needed.

With the increase of oxidant concentration, platelet aggregation increased in healthy controls and the CHF subgroups. It is worth noting that the increase of platelet aggregation in samples with the medium oxidant concentration (ADP2 0.0778 mmol/L) was not reliable in CHF and healthy controls. Both in CHF patients and healthy controls, platelet aggregation with ADP was determined to be significantly higher in samples with the highest concentration of oxidant (ADP1 0.15 mmol/L), compared to the samples with the lowest oxidant concentration (ADP3 0.0389 mmol/L) (*p* = 0.02 and *p* = 0.045 in healthy controls and in CHF patients, respectively). Plasma antioxidative enzyme activity was decreased in patients with heart failure (HF) [24]. An increase in lipid peroxidation markers and an encroachment of antioxidant reserves in HF have been reported in humans as well [25]. Platelets represent very sensitive sentinels in the blood that are known to be activated by oxidative stress and inflammation [7]. It has been shown that low high density lipoprotein (HDL) concentrations relate to endothelial dysfunction [26] and increase low density lipoprotein (LDL) oxidation. HDL enhances the synthesis and bioavailability of nitric oxide and possess anti-oxidative as well as anti-inflammatory properties, in part by modulating the function of various cell types that are involved in the thrombosis process [7]. It has been found that hypochlorite-oxidized LDL significantly increased the platelet adhesion to collagen and the formation of platelet–platelet aggregates [7], which was in principal agreement with findings obtained with trace metal-oxidized LDL [27]. However, consequences of platelet–leukocyte interaction are complex, as activated leukocytes enhance platelet aggregation and thromboxane release [28], while binding of activated platelets to resting leukocytes results in leukocyte activation [7]. The reason why oxidant enhancement of platelet aggregation in healthy controls, where antioxidative potency is expected to be sufficient, was not lower compared to CHF patients remains unrevealed. It may be suggested that if CHF patients had not been receiving standard treatment, a more intense platelet aggregation might have been induced in this group by an oxidant compared to healthy controls.

Platelet aggregation differences among the CHF subgroups in our study may partially be explained by different dityrosine concentrations. Oxidative modifications of proteins are conversion of tyrosine residues to 3-chlorotyrosine, 3-nitrotyrosine, dityrosine, or methionine oxidation [29]. Oxidative modifications of fibrinogen chains may alter its normal function [30]. Furthermore, fibrinogen isolated from postmiocardial infarction plasma with high total protein carbonyls gave a significantly higher mean rate of aggregation compared with low carbonyl plasma [2]. We found the highest dityrosine concentration in NYHA IV subgroup and it decreased gradually with the decrease of oxidant concentration. These results are comparable with other studies. The plasma levels of nitrotyrosine, a protein marker of nitric oxide-derived oxidants, are enhanced in the plasma of coronary artery disease patients and independently predict cardiovascular risk [15]. It has been found that HDL, oxidatively modified by the oxidant hypochlorous acid occurring in vivo, shows specific and saturable binding to human platelets. Hyp-OxHDL triggers platelet aggregation and causes a dose dependent expression of CD62P and CD40 ligand on the platelet surface. The occurrence of these ligands is associated with higher rates of platelet activity [8]. Badrnya et al. [7] found that hyp-OxLDL significantly increased platelet adhesion to collagen and the formation of platelet–platelet aggregates. Our observation of significantly higher dityrosine concentration in the CHF group compared to healthy controls confirmed lower antioxidant potency in CHF. Besides, we obtained a weak negative correlation between spontaneous aggregation without the oxidant and dityrosine concentration. The increase of oxidant concentration resulting in raising dityrosine concentration confirms that with the increase of oxidant concentration, the antioxidative potency is diminishing. This is consistent with the finding [8] that oxidative stress results from the exhaustion of antioxidative capacity, which leads to the activation of platelets.

Limitations of this study include a small sample size, which included only patients with reduced left ventricular ejection fraction. Additionally, we did not measure other biomarkers of prothrombotic state (e.g., D-dimers) or even more specific platelet activation markers (e.g., flow cytometric expression of P-selectin (CD62P)), or the activated conformation of glycoprotein (GP) IIb/IIIa (PAC-1) binding. Specific markers investigation could possibly explain the decrease of platelet aggregation in NYHA IV subgroup compared to NYHA III subgroup. Indeed, our aim was to evaluate the influence of the oxidant HOCl on platelet aggregation and dityrosine concentration in CHF patients and healthy controls and find correlations between the measured readings.

Some hypothesis on our findings can by drawn. The fact that dityrosine concentration was higher in the CHF group and correlated with the CHF functional class, as well as the higher platelet aggregation in NYHA III subgroup compared to NYHA II subgroup, proves that human platelets activated by oxidative stress lead to prothrombotic state. Significantly higher dityrosine concentrations in CHF group represent oxidative stress. Oxidants might be one of the reasons for thrombotic complications in CHF with dityrosine concentration being representative of the prothrombotic state. The greater dityrosine concentration in CHF patients and its correlation with the severity of the disease are interesting and open a new area of investigation if antioxidants can be useful for avoiding thrombotic complications in CHF. Large studies appear to be needed in this area to better understand the pathophysiological ramifications of these observations, which would help in the development of better therapeutic targets.

## 5. Conclusions

In conclusion, the effect of higher concentration of oxidant is similar in both groups. In conclusion, the effect of higher concentration of oxidant is similar in both groups. Having estimated the differences of various concentrations of oxidant HOCl impact on platelet aggregation in CHF, we found that 0.15 mM/L and 0.0389 nM/L oxidant concentrations enhance platelet aggregation both in CHF patients and healthy controls. Higher dityrosine concentrations were observed in CHF group and increased with CHF severity (depending on the functional CHF class). We found that the intensity of spontaneous aggregation with oxidant added was weakly associated with the CHF functional class too. Platelet aggregation with ADP in samples with oxidant added correlated with LV EF.

Our study was the first to evaluate platelet aggregation and dityrosine concentration in CHF subgroups according to NYHA functional class.

## Figures and Tables

**Figure 1 medicina-55-00198-f001:**
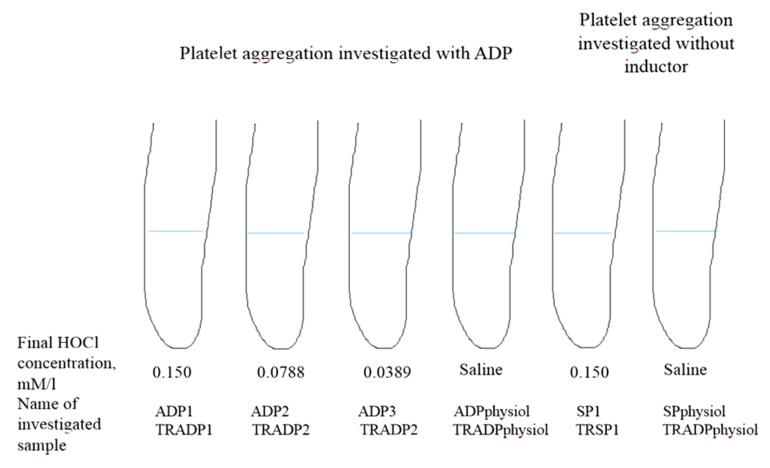
A scheme presenting the sample preparation and protocol of aggregometric and dityrosine concentration studies. TR, dityrosine concentration; ADP, adenosine diphosphate; SP, spontaneous aggregation (without inductor); 1,2,3 different concentrations of hypochlorous acid (HOCl) added.

**Figure 2 medicina-55-00198-f002:**
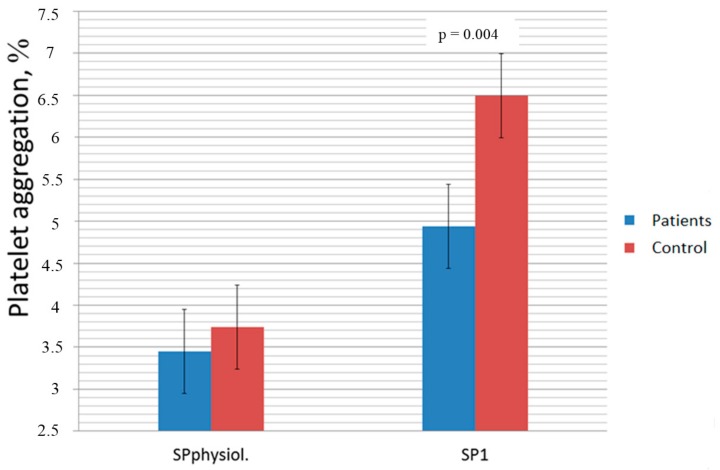
Differences of spontaneous aggregation in chronic heart failure (CHF) patients and healthy controls with and without oxidant added. SP, spontaneous aggregation; 1, highest oxidant concentration added; physiol, saline added.

**Table 1 medicina-55-00198-t001:** Clinical characteristics of patients and control subjects (A flt/A fib, atrial flutter/fibrillation).

Clinical Variables	II NYHA (*n* = 21)	III NYHA (*n* = 24)	IV NYHA (*n* = 22)	Healthy Controls (*n* = 31)	*p*-Value
Age, years, mean ± SD	65.19 ± 13.23^‡^	65.0 ± 13.24^‡^	61.39 ± 13.57	55.42 ± 8.88	0.024
Left ventricular ejection fraction, %, mean ± SD	44.19 ± 5.81^ǂ^	39.00 ± 10.86*	26.50 ± 13.73	56.94 ± 3.08^‡^	<0.001
6-min walking test, m, mean ± SD	334 ± 41^ǂ^	324 ± 39^+^	222 ± 79*	573 ± 85^‡^	<0.001
Platelet count, × 10^9^/L, mean ± SD	199 ± 54.21	199.38 ± 47.30	208.44 ± 61.33	217.61 ± 47.7	0.033
Platelet volume, fl, mean ± SD	10.57 ± 1.1	11.24 ± 1.03	11.03 ± 1.0	8.7 ± 1.01^‡^	<0.001
Diuretics, *n*	9	18	17	0	
Beta-blockers, *n*	16	21	11	0	
ACE-inhibitors, *n*	15	21	14	0	
Nitrates, *n*	5	11	11	0	
Digoxin, *n*	1	3	4	0	
Statines, *n*	12	11	7	0	
Heparine, *n*	6	11	11	0	
Calcium channel blockers, *n*	1	0	1	0	
Thrombosis, *n*	0	0	3	0	
Aflt/Afib, *n*	7	14	13	0	
Renal failure, *n*	2	5	12	0	
Obesity, *n*	11	10	11	0	
Dyslipidemia, *n*	11	10	9	0	
Cardiovascular diseases, *n*	10	11	12	0	

* Statistically significant difference between NYHA IV and NYHA III, ^ǂ^ statistically significant difference between NYHA II and NYHA IV, + statistically significant difference between NYHA II and NYHA III, and ^‡^ statistically significant difference between CHF patients and healthy controls.

**Table 2 medicina-55-00198-t002:** Platelet aggregation changes with different concentrations of oxidant in CHF patients and healthy controls.

Used Oxidant Concentration, mmol/L	CHF Patients (*n* = 67)			Healthy Controls (*n* = 31)		
	Platelet Aggregation, Induced with ADP (%, mean ± SD)	Increase of Platelet Aggregation in Samples with Oxidant, %	*p*-Value	Platelet Aggregation, Induced with ADP (%, Mean ± SD)	Increase of Platelet Aggregation in Samples with Oxidant, %	*p*-Value
Saline	57.52 ± 19.77			66.3 ± 9.63		
0.0389 (III)	66.55 ± 21.56	15	<0.001	78.94 ± 14.31	14	0.005
0.0778 (II)	58.64 ± 18.54	2	0.74	67.16 ± 10.96	2	0.45
0.15 (1)	69.10 ± 21.06	19	<0.001	81.61 ± 13.26	20	0.005

**Table 3 medicina-55-00198-t003:** Dityrosine concentration in CHF and healthy control group samples with different oxidant HOCl concentrations.

Used Oxidant Concentration, mmol/L or Saline	Dityrosine Concentration in Plasma (Relative Units of Fluorescence) (Mean ± SD)		*p*-Value
	CHF Patients (*n* = 67)	Healthy Controls (*n* = 31)	
Saline	1.54 ± 0.48	1.27 ± 0.53	0.049
0.0389 (III)	1.56 ± 0.49	1.34 ± 0.40	0.032
0.0778 (II)	1.55 ± 0.51	1.55 ± 0.38	0.310
0.15 (1)	1.73 ± 0.54	1.82 ± 0.48	0.025

**Table 4 medicina-55-00198-t004:** Platelet aggregation differences in CHF subgroups.

Used Oxidant Concentration, mmol/L or Saline	Platelet Aggregation, Induced with ADP in Samples with Oxidant or Saline Added (mean ± SD)			Reliability of Data Differences between Samples of Patients Subgroups (*p*-Value)		
	NYHA II	NYHA III	NYHA IV	NYHA II and NNYHA III	NYHA III and NYHA IV	NYHA II and NYHA IV
Saline	56.13 ± 12.57	61.75 ± 22.44	53.11 ± 21.04	0.001	NS	0.056
0.0389 (III)	64.88 ± 12.36	71.5 ± 23.49	61.44 ± 24.75	<0.001	NS	0.012
0.0778 (II)	55.69 ± 11.22	64.21 ± 20.44	53.83 ± 20.0	<0.001	NS	0.028
0.15 (1)	66.81 ± 11.79	74.58 ± 23.31	63.83 ± 23.52	<0.001	NS	0.012

NS, statistically insignificant.

**Table 5 medicina-55-00198-t005:** Dityrosine concentration in CHF subgroup samples with different oxidant HOCl concentrations.

Used Oxidant Concentration, mmol/L or Saline	Dityrosine Concentration in Plasma with Oxidant or Saline Added (mean ± SD)			Reliability of Data Differences between Samples of Patients Subgroups (*p*-Value)		
	NYHA II	NYHA III	NYHA IV	NYHA II and NNYHA III	NYHA III and NYHA IV	NYHA II and NYHA IV
Saline	1.50 ± 0.51	1.47 ± 0.49	1.71 ± 0.46	NS	0.237	0.037
0.0389 (III)	1.56 ± 0.51	1.42 ± 0.40	1.63 ± 0.52	NS	0.260	0.680
0.0778 (II)	1.45 ± 0.39	1.38 ± 0.44	1.09 ± 0.65	NS	0.023	0.048
0.15 (1)	1.67 ± 0.57	1.58 ± 0.58	2.11 ± 0.43	NS	0.014	0.066

NS, statistically insignificant.

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
