# Peer review of "The Effect of Oxidant Hypochlorous Acid on Platelet Aggregation and Dityrosine Concentration in Chronic Heart Failure Patients and Healthy Controls"

_medicina, 2019, doi:10.3390/medicina55050198_

Reviewer 1 Report

Dear Authors,

Your manuscript entitled “The effect of Oxidant Hypochlorous Acid on Platelet Aggregation and Dityrosine Concentration in Chronic Heart Failure Patients and Healthy Controls” aims to describe the effect of oxidant HOCl on platelet aggregation and dityrosine concentration in CHF patients and healthy controls. The idea of this project is interesting considering the importance to study some specific disease such as chronic heart failure and its complications. The study of platelet aggregation in this clinical condition can be of relevant interest to choose adequate therapies or modulate particular therapeutical protocols just used.  

In the complex, the manuscript could give an interesting contribute about the possible role of hypochlorous acid on platelet aggregation induced by ADP. The experimental design is interesting and well documented and described. Some question are on figure 1 where platelet aggregation (%) is indicated with numbers from 2.5 to 7.5. Normally platelet aggregation graph is represente with a scale from 0 to 100%. The authors should explain this.

In addition, discussion should be reduced and focused particularly on the aspects of the paper. In some point references should be added.

On these basis, the manuscript can be accepted after minor revisions.

Author Response

1.      Fig. 1 represents spontaneous aggregation. It is investigated without any inductor and rarely exceedes 20%. In Fig. 1 platelet aggregation is indicated with numbers from 2,5 to 7,5 % to more explicitly demonstrate the differences between investigated groups and samples.

2.      Discussion section has been reduced.

Reviewer 2 Report

Although interesting the paper suffers of some limits that has been clearly stated by the authors in the limitation secion i.e the low number of subjects and the absence of other dosage of molecules implicated into the thrombosys process.

Beside this both results and conclusion are not well written and a little confounding and need to be re-written. Furthermore conclusion are inconclusive particularlly regarding the principal finding. As an example one of the main finding (aggregation is increased in NYHA III but reduced in NYHA IV) is not well discussed and no hypothesis on how this works is done.

Results an conclusion should be rewritten focusing on the main finding, explaing it in a better way and drawing hypotesis on these findings.

Author Response

1.      Limitations of study are presented in separate section.  The small number of subject and absence of other dosages of oxidant and other molecules implicated into the thrombosys process were chosen because this was only a pilot study. The obtained results proving oxidant effects on platelets and increasing dityrosine concentration according to NYHA functional classes would allow to plan more extensive analysis.

2.      Conclusions have been rewritten acording to the findings.

3.      We do not have and do not find in the literature any findings explaining platelet aggregation differences between NYHA III and NYHA IV subgroups. Maybe future investigation, involving specific platelet activation markers and a bigger number of subjects could help to explain this fact.

4.      Hypotesis based on papers findings have been drawn et the end of the paper.

Reviewer 3 Report

The aim of the study was to assess the impact of oxidant agents (in particular HOCl) on platelet aggregation and dityrosine concentration in patients with chronic heart failure compared with healty subjects and to evaluate platelet aggregation and dityrosine concentration among patients in different NYHA functional class.

We must congratulate the Authors for their intuition. The effects of oxidant agents on platelet aggregation could be further investigate taking also into account clinical data, opening the field for new specific therapeutic approaches.

The Authors give an interesting and argued interpretation of their results, correctly identifying the main limitations of their study such as the small sample size including only patients with LV reduced ejection fraction, not considering patients with heart failure with preserved and mid-range ejection fraction, the absence of the analysis of other biomarkers of prothrombotic state for a more precise stratification of prothrombotic risk of the various subgroups analyzed and of ulterior and more specific platelet activation markers.

However, I have some points to address to the Authors:

- In the Introduction the Authors could better explain the choice of dityrosine concentration as a marker of oxidative stress.

- The section Material and methods should be reported after the section Introduction.

- In the section Material and methods a more precise description of patients selection, including the number of patients screened not meeting the inclusion criteria for the study and the description of healthy controls selection, should be included.

- At page 9, line 326: "antiaggregant " instead of "anitagregant"

- In the section Results the analysis of the data is difficult to understand for the reader and it should be presented in a clearer and more consistent way. 

- In table 1, the p value for MPV, nitrates and other medications and comorbidities is missing. 

- In table 3 the p value of dityrosine concentration in the samples with saline is different from that reported at page 4 line 101 (0.045 vs 0.049).

- In table 5 "platelet aggregation, induced with ADP ..." should be replaced by "Dityrosine concentration in plasma..."

- At page 5, line 157-160: it is not clear the correlation with LV EF: platelet aggregation was differently correlated with the reduction of LV EF in the different samples?

- In the section Discussion at page 6, line 180-181: the statement is not perfectly supported by what is written in the section Results since no comparison between healthy controls and CHF subgroups is shown.

Author Response

1.      In the Introduction the choise of dityrosine has been explained.

2.      The manuscript sections in our paper follow the requirements of Medicine Instructions for Authors: Introduction, Results, Discussion, Materials and Methods, Conclusions, Back Matter.

3.      The description of healthy controls selection has been included in the section Material and methods. The number of patients screened not meeting the inclusion criteria for the study has been omited because it did not influence the results.

4.      The mistake has been corrected: antiaggregant.

5.      The section Results has been revised and presented in as clearea way as possible.

6.      Below Table 1 is written that the differences in medication used and comorbidities in CHF group were not statistically significant. So, to avoid overloading they are omitted in the table.

7.      The mistake has been corrected: in table 3 has been replaced by 0,049.

8.      The mistake has been corrected: in table 5 incorrect name has been replaced by correct.

9.      According our findings, the correlation strenth of LVEF with platelet aggregation depends on oxidant concentration in the plasma samples.

10.   The mistake has been corrected. The missleading sentence has been rewriten as:“...reveald significantly lower...in CHF group compared to ...“.

Round  2

Reviewer 2 Report

Paper little improves after changes but if editor agree can now been accepted

Author Response

The spell check has been done.

Reviewer 3 Report

In this new version of the paper the Authors have corrected the mistakes reported in the previous version and, in particular, the new Results section is clearer, smoother and easier to read and a description of healthy controls has been added.

I have only two minor corrections to suggest: 

- At page 9, line 327: "enrolled" instead of "enroled".

- At page 8, the sentences  from line 295 to line 315 should be presented in the paragraph Discussion and not in the section Limitations. 

Author Response

1.      The mistake at page 9, line 327 "enroled" has been corrected to "enrolled" .

2.      The part of the text from Limitation section has been moved to the end of the Discussion section.